# Compression Performance Analysis of Experimental Holographic Data Coding Systems

**DOI:** 10.3390/s23187684

**Published:** 2023-09-06

**Authors:** Tianyu Dong, Kwan-Jung Oh, Joongki Park, Euee S. Jang

**Affiliations:** 1Department of Computer Science, Hanyang University, Seoul 04763, Republic of Korea; tianyu@hanyang.ac.kr; 2Digital Holography Research Section, Electronics and Telecommunications Research Institute, Daejeon 34129, Republic of Korea; kjoh@etri.re.kr (K.-J.O.); jkp@etri.re.kr (J.P.)

**Keywords:** holography, data compression, image communication

## Abstract

It is challenging to find a proper way to compress computer-generated holography (CGH) data owing to their huge data requirements and characteristics. This study proposes CGH data coding systems with high-efficiency video coding (HEVC), three-dimensional extensions of HEVC (3D-HEVC), and video-based point cloud compression (V-PCC) codecs. In the proposed system, we implemented a procedure for codec usage and format conversion and evaluated the objective and subjective results to analyze the performance of the three coding systems. We discuss the relative advantages and disadvantages of the three coding systems with respect to their coding efficiency and reconstruction results. Our analysis concluded that 3D-HEVC and V-PCC are potential solutions for compressing red, green, blue, and depth (RGBD)-sourced CGH data.

## 1. Introduction

The rapidly growing interest in computer-generated holography (CGH) is manifested in its broad range of applications, including security holograms and volumetric art displays. Typically, holography requires a single laser for object irradiation, which makes it difficult to record a hologram scene with depth, parallax, and other properties in real-world situations. With the increasing popularity of augmented reality (AR)/virtual reality (VR) applications, the demand for three-dimensional (3D) content with high resolution and quality is continuously increasing. Current CGH methods can be classified into two categories: wavefront- and ray-based CGH methods [1]. Wavefront-based CGH methods generate three-dimensional (3D) scenes and objects using point cloud, polygon, and layer-based models. Owing to the independence between light primitives, acceleration using parallel-computed light primitives offers higher generation speeds. Red, green, blue, and depth (RGBD)-based CGH generation also provides a simple method used to produce CGH content that can be used in AR/VR applications. The acquisition of color images and 3D depth is easy to achieve using existing 3D models or commercial hardware equipment [2]. However, while implementing CGH applications, the transmission of raw holographic data remains a longstanding challenge. Research shows that streaming holographic videos at 60 frames per second over a network requires a bandwidth between 100 Gbps and 1 Tbps [3]. This indicates that it is difficult to store and transfer holography data in current network environments. When using 3D model data, this task becomes simple: existing media codecs and their standards can provide efficient transmission solutions.

Some related studies [4,5,6] employed JPEG, JPEG 2000, and high-efficiency video coding (HEVC) to compress CGH data. Compression of RGBD data using JPEG has also been studied [4]. At a bit rate of 6.5 kB/s, a 320 × 240 sized RGBD data sample was evaluated. However, the comparison results showed that an RGBD-based image compressed using JPEG yielded a better compression performance than a hologram-based image compressed using JPEG for numerical reconstruction (NR). This is because hologram-based images contain sharp edges and complex patterns that make it difficult to perform JPEG compression with high efficiency. In [5], it was shown that the compression of phase-shifting digital holography using JPEG 2000 yielded an acceptable reconstruction quality compared to JPEG owing to the better performance of JPEG 2000 in encoding complex patterns. In [6], a comparison of the compression efficiency between HEVC and JPEG 2000 was presented. As a result of the unique characteristics of the amplitude phase data format, an undesirable quality of the reconstructed image was observed at low bitrates.

Other related studies [7,8,9,10] focused on compressing the hologram pattern, which is different to natural images. In [7], an integer Fresnel-transform-based JPEG 2000 compression method archived 0.12–2.83 bits per channel of bitrate savings in lossless conditions. On the other hand, transform-based methods such as the Ramanujan-periodic-transformed hologram data with fpzip [8] or overcomplete Gabor’s dictionary matching pursuit [9] achieved higher lossless compression ratios. In [10], a wave atom coding method was proposed to compress Fourier or defocused content in the macroscopic holograms, and it shows better performance than JPEG 2000 and equivalent performance of HEVC.

A recent work [11] has shown that the shearlet transform can be used for light field reconstruction, and this could also benefit the processing of holographic images. Other related work, such as a low complexity HEVC encoder [12], could benefit the system by reducing the encoding time and complexity of the H.265/HEVC encoder used for hologram data.

More complex methods such as neural networks and deep learning have also been tested [13,14]. In [13], a deep neural network was used to improve the reconstruction quality method, showing a better performance than JPEG 2000 and HEVC. Another approach [14] combined a deep convolutional neural network and JPEG to reduce the artifacts at high frequency and provide better quality than the JPEG compressed result.

These compression studies are focused on hologram data, as shown in Table 1. Lossless compression methods deal with hologram data patterns by using complex algorithms. Several codecs were used to compress the input data of the CGH. It is apparent that the lossy compression of the input data for CGH has a direct impact on the reconstruction quality. Furthermore, the impact of lossy compression on the CGH quality distortion has not been systematically discussed.

To evaluate the codec for the source media for CGH generation, it is not only important to compare the performance in terms of the compression rate, but also to analyze the rate distortion (RD) performance for the CGH source input and output. In addition, it is crucial to compare the numerically rendered CGH quality based on RD performance and objective reviews. The merits and demerits of each codec can be determined using this process.

In this study, we provide a comparative analysis of several legacy media codecs (i.e., high efficiency video coding (HEVC), 3D extensions of HEVC (3D-HEVC), and video-based point cloud compression (V-PCC)) based on their compression efficiency and rendering quality. The analysis was based on an objective review of the reconstruction results and a subjective metric analysis of the final and intermediate results. The encoding procedures and strategies for adapting each codec are also discussed. The main contribution of this paper lies in the fact that we provided a framework-based test bed for holographic data compression. We provided a minimal set of test data to prove the concept of comparison of different codecs on hologram.

The remainder of this study is organized as follows. In Section 2, we analyze the advantages of using RGBD and a point cloud as source inputs for the CGH. We also provide an overview of the structure of HEVC, 3D-HEVC, and V-PCC used in the proposed experiment. In Section 3, we describe the procedure for applying each codec scheme in detail, including color space conversion, RGBD–point cloud conversion, and encoder configuration. The defects in the numerical reconstruction caused by each lossy compression method and their possible fixes were also discussed. The evaluation process and experimental settings are presented in Section 4. In Section 5, we evaluate the compression rate and the subjective and objective numerical reconstruction rendering quality statistics for each codec.

## 2. Background

A simple method for recording a 3D scene is to use the RGBD format, which transforms a 3D surface into a depth map and intensity information or records a scene with depth information and the corresponding color. All these images share the same pixel resolution for depth and RGB. The use of RGBD to generate a hologram scene involves the following procedure: First, a 3D object must be rotated to create multiple views in the general case, wherein a multiview hologram is generated. Instead of object irradiation, we can consider it as a self-illuminating object; that is, the light source comes from it. Second, for each depth, it is necessary to calculate the recording planes, and then use these to generate holographic planes using the Fourier transform. Fresnel diffraction is used to compute the optical field on the hologram plane, and the resulting data are a set of complex numbers [2].

While a pair of RGBD images can be used to record only a single directional view of the 3D scene, point cloud data can provide an omnidirectional representation of multiple viewpoints. Using a set of points with 3D coordinates and color information, a point cloud can represent a high-precision 3D object. A 3D model represented as a point cloud can record other attributes of an object, such as reflectance or transparency, which will enrich the perceptible effect of the object. In addition to attribute recording, a computer graphic processing unit (GPU) can be used to speed up CGH generation by dividing the point cloud data into small batches [15]. The conversion between RGBD and point cloud data is relatively simple using an orthographic projection.

### 2.1. HEVC, 3D-HEVC, and V-PCC

We used HEVC [16], 3D-HEVC [17], and V-PCC [18] to compress RGBD images and point clouds. These are all international coding standards from MPEG. HEVC was developed in 2013 and achieved a 30–67% bitrate reduction compared to its predecessor advanced video coding (AVC) for the same level of video quality [16]. Compared with AVC, HEVC includes several advanced coding tools, one of which is a new partition method called coding tree unit (CTU). Using the CTU structure, HEVC allows image blocks to easily find their best prediction blocks and most efficient transform shapes. Both the inter-frame and intra-frame predictions were designed with more candidates for accurate predictions. The extended color space and bit depth were supported. Advanced profiles and bitrate levels were also enhanced to support a high image resolution [19]. Figure 1 shows the coding structure of HEVC and the relationship between the coding tools.

A 3D extension of HEVC (3D-HEVC) was developed based on HEVC and multiview HEVC and provides an increased coding efficiency by combining the coding of texture and depth for 3D displays. In 3D-HEVC, several new coding tools have been introduced for multilayer coding. Inter-layer prediction can improve the performance when encoding the RGB and depth layers. Depth intra-image prediction achieved a better performance than HEVC version 1 on low-complexity depth information, and inter-view illumination compensation improved the coding quality when encoding different view angles. Other tools include view synthesis prediction, which predicts the texture and depth information of a view from other views; inter-view (IV) motion prediction, which uses motion vectors from other views to predict the motion of a current view; and advanced residual prediction, which exploits the correlation of sample prediction errors in different views or auxiliary pictures [17]. Additionally, adaptive weighting of an IV sample prediction has been enabled by illumination compensation, and depth-based block partitioning combines two predictions for texture coding according to a sub-partitioning derived from a corresponding depth block. Overall, these tools help to achieve a higher compression efficiency and reduce the bit rate in the encoding process.

Starting from the 3D Graphic Group in MPEG, point cloud compression (PCC) has been developed with a focus on the compression of the three categories of point cloud data. In the common test conditions of MPEG PCC [20], the three categories of point cloud data are static, dynamic, and dynamic acquisition point clouds. Categories one and three are compressed using geometry-based PCC (G-PCC), and category two is compressed using video-based PCC (V-PCC) [18]. The V-PCC encoder employs 3D-to-two-dimensional (2D) patch generation and arranges these patches into a sequence of images. Patch generation is one of the main functional elements of the V-PCC codec architecture. It refers to the process of dividing a point cloud into smaller patches, which are then projected onto a 2D plane. The patches are ordered by their size, and their location is determined by several non-normative methods. Normal estimation and segmentation are important and time-consuming steps in the point cloud compression process. Normal estimation is the process of calculating the surface normal vectors for each point in the point cloud. These normal vectors are used to represent the orientation of the surface at each point. Segmentation is the process of dividing a point cloud into smaller regions or segments based on certain criteria. These two steps determine the images to be coded in video codecs. Then, a video codec is used to compress the occupancy map, geometry, and texture images separately. Figure 2 shows a block diagram of the V-PCC encoding scheme.

### 2.2. Color Space Conversion

Several issues must be considered when selecting a codec for hologram source compression. The first is the image size. Codecs are typically designed to satisfy the requirements of different display devices with various pixel resolutions. Thus, pixel resolution is supported by different quality levels. Second, these codecs were designed to fit a designated computational complexity such that only a limited color depth (i.e., 8 or 10 bits) and subsampling methods (i.e., YUV 4:4:4 or YUV 4:2:0) were supported at a constant value.

As RGB is a popular choice in 3D graphic color space, data loss may occur owing to the color space conversion between RGB and YUV. Thus, using the latest HEVC with range extension (HEVC RExt) [21], YUV 4:4:4 pixel sampling can provide full-color space support. The 3D-HEVC was designed for color and depth view image compression, and it can use the correlation information between the color and depth layers by inter-layer prediction, which is an ideal solution for RGBD data. However, the color space profile support of the 3D-HEVC test model (3DVC) is limited to HEVC version 1, which does not include RExt support. Thus, all RGB color values must be stored in the YUV 4:2:0 pixel subsampling format using 3D-HEVC, which causes chrominance component conversion loss. The V-PCC for 3D model compression follows the configuration of the color space support in the video encoder. Furthermore, the color space conversion also depends on the compression demands when designing the V-PCC encoder.

## 3. Design of Coding Systems

In this study, we evaluated three different coding systems to compress the CGH generation sources. The coding system includes a 2D format converter, coding tools, a point cloud converter, a CGH generator, and a renderer. Figure 3 shows three unique designs using three different codecs and their procedures. The 2D format converter converts the color space of the image data from RGB to YUV. The coding tools selected for these three designs were the HEVC, 3D-HEVC, and V-PCC codecs. The test model for each codec is recommended by the International Organization for Standardization (ISO) and/or the International Telecommunications Union (ITU). The point cloud to RGBD converter was designed using an orthogonal projection on 3D points and 2D pixel values. RGB and depth images share the same 2D coordinates, with color information stored separately from the *z*-axis. The CGH generator uses RGBD images as inputs and generates real-imagery-formatted CGH files.

The 3D-HEVC-based coding system uses a 3D-HEVC for RGBD compression, as shown in Figure 3a. In this method, a 2D format converter is required before and after the coding process for the conversion between the RGB and YUV color space. This system was designed to exploit the correlation between RGB and depth data. In addition, the system reduces other conversion errors of the CGH source, such as geometric distortion in the V-PCC coding system and lightwave data distortion in the HEVC coding system. Furthermore, 3D-HEVC supports multiview RGBD images, which means that it can obtain coding benefits from inter-multiview prediction. It also provides extensibility for complex multiview RGBD-based hologram source data.

The second method employs HEVC to compress the real imagery CGH format, as shown in Figure 3b. We used this method for comparison because the CGH itself (and not the source media) is compressed. The generated CGH is in the form of a complex hologram and contains imaginary and real number parts. Additionally, we used a random phase image on the encoder and decoder sides of the entire process. It should be noted that the input and output formats are YUV. In this system, a 2D format converter was used to convert the BMP format to the YUV format. The use of HEVC deployed in a variety of devices can reduce implementation costs. In addition, full YUV 4:4:4 color support can reduce the color space conversion loss.

The point cloud method shown in Figure 3c uses V-PCC for compression. Point cloud conversion is necessary for this process. Compared to the 3D-HEVC method, V-PCC can represent a wider range (e.g., 12 or 16 bits) of depth information and a more complex geometry structure. However, the complexity of point cloud coding is higher than that of the two previous systems because of the complex patch generation process at the encoder. Even on the decoder side, the point cloud reconstruction and smoothing processes are very time consuming.

## 4. Performance Evaluation

This section discusses the evaluation details, including experimental data, different encoding options for the three systems, data conversion tools, and evaluation processes. These configurations affect the reconstruction quality in multiple dimensions.

### 4.1. Experimental Data

Three sets of single-view static images were selected for this experiment [17]. Mario, Cube, and Pororo show the experimental data, as shown in Figure 4. The depth of each image is limited to the range of 0–255. The corresponding color for each depth point was represented in an 8-bit RGB color space data range. Furthermore, test data were generated at different image sizes to evaluate the performance at different resolutions. The image width and height of the Mario sequence were 1536 × 1536 pixels, and those of the Cube and Pororo sequences were 768 × 768 pixels.

### 4.2. 3D-HEVC

A 3D-HEVC test model (3DVC) (version 16.3) was used in this implementation [22]. As a result of 3D-HEVC being developed for multiview television, a single view with color and depth is not supported. Thus, because 3D-HEVC was developed from multiview-HEVC (MV-HEVC), it does not take a single pair of RGBD images as input. For this reason, we used three copies of the RGBD data from three different camera sources as inputs for the non-common test condition (CTC) three-view encoding configuration. The system converts RGBD data into YUV 4:2:0 format as input to the 3D-HEVC. Subsequently, we used one view as the center, and the other two views had a 10^−7^ degree difference.

During the performance evaluation, the all-intra encoding configuration was selected such that only one frame for each hologram image was encoded. As the final analysis needs to be extinguished from a standard perspective, we chose the rate points used from standardization period of each codec as the demand for each codec is different, and the QP value indicates different choice of coding tools. The 3D-HEVC CTC [23] required five bitrate points to be called, from R1 to R5, and we extended the rate to obtain a higher quality point for comparison with other codecs. Table 2 lists the rate point and quality parameter (QPs).

In the standardization process, the geometry QP is usually set with a lower QP value than the color QP to maintain the 3D structure quality. This is because the geometry information is crucial for 3D graphic coding, and any loss of geometry information can result in a significant degradation of the overall quality of the 3D graphics. Therefore, to maintain the quality of the 3D structure, the geometry QP is set lower than the color QP.

### 4.3. HEVC

The HEVC test model (HM) (version 16.20) was used in this system [24], and an all-intra encoding configuration was selected [23]. While evaluating the coding efficiency of HEVC, four different operation points were used, and two lower rate points were added for the evaluation. All the QP values for each rate point are listed in Table 3. YUV 4:4:4 was selected to store the real/imaginary CGH data.

As we mentioned, the HEVC system compressed the hologram data itself, and the real/imaginary parts are shown in Figure 5; as indicated, the complexity of compressing real/imaginary images may differ when this method is used compared with the other 3D-HEVC and V-PCC methods.

### 4.4. V-PCC

A V-PCC test model (TMC2) (version 7.0) was selected for this system [25]. An all-intra configuration was selected for the encoding process. It should be noted that in the CTC [20], the all-intra configuration uses YUV 4:2:0 for the embedded HEVC setting, and images at the near-end and far-end layer are encoded with one I frame and one P frame. Furthermore, the occupancy map precision was set to two for the higher rate point R5, and four for R1–R4. In Table 4, we also extended the rate to obtain a higher quality point for V-PCC to compare with other codecs. An occupancy map in V-PCC is a 2D representation that indicates whether a given position (block) in the map is occupied by patched points from the 3D point cloud or not. The precision of the occupancy map defines the block size.

The size of the patched image is essential for coding efficiency. The minimum image patch size in the encoder defines the initial size of the video. We used the width and height of the RGBD image; the images in the Mario sequence had sizes equal to 1536 × 1536, and those in the Cube and Pororo sequences had sizes equal to 768 × 768.

### 4.5. Other Tools

The problem of color space conversion loss has already been discussed in depth by standardization organizations, such as MPEG, which study the compression of various media data such as images, videos, textures, and depth. As an extension of this discussion, we also decided on the format in our proposed system. FFmpeg version 4.2 [26] was selected as a 2D format converter to conduct the color space transformation based on ITU-R Recommendation BT.709 [27]. The format conversion includes the standard ITU-R Recommendations BT.709 and BT.2020, which are also used by MPEG. We followed this practice as well. The input RGB color space images were stored in BMP format and the YUV color space images were stored as raw YUV data files.

We designed a point cloud converter in MATLAB (version 2019b) [28] to convert the point cloud and RGBD images. The horizontal and vertical positions of the pixel used in the 2D images are the *x*- and *y*-axis coordinates in the 3D coordinate system, respectively, and the depth information was used as a *z*-axis coordinate. The color information then followed the projection of the depth position. The converter constructed a point cloud PLY file using the header and data components. The header contained a description of the point cloud, such as the number of points and data types for the given attributes. The rendered images of the converted point cloud data are shown in Figure 6.

The rendering process uses a spatial light modulator to accomplish reconstruction [4]. The renderer uses amplitude and phase data to modulate light waves in space and time to reconstruct the hologram. In this design, we let the CGH renderer [4] use the same phase file for a simplified rendering process, because the phase data do not interfere with the rendering process.

The NR results were generated from the hologram CGH at a specific depth and viewpoint. Instead of generating all the NR results of the test dataset at each depth (between 0 and 255) and with a fixed viewpoint in the front center for full depth range evaluation, we selected three depths for each sequence to evaluate the quality near, far, and medium depths. To better compare the subjective image quality, we also generated NR images at the intermediate depth of the far, middle, and near scenes. The NR images were generated using the depth data presented in Table 5.

### 4.6. Evaluation Process

For the evaluation, we analyzed the NR results of the three systems and then performed subjective and objective evaluations. Subjective evaluations focused on color restoration and object edge distortion. The objective evaluation analyzed the peak signal-to-noise ratio (PSNR) and structure similarity (SSIM) of the final NR images generated from the 3D-HEVC, HEVC, and V-PCC coding systems. When evaluating objective performance, referring to the Bjontegaard delta bitrate (BD-BR) performance evaluation method [29], if the curves of each codec cover each other, the performance difference can be evaluated by the area between the curves. This is also the reason why we want to increase the rate points for each codec.

We also evaluated the PSNR and SSIM on intermediate RGBD results in the 3D-HEVC coding system, and the point cloud error (PC_Error) results in the V-PCC coding system. The PC_Error is described by the following equation:(1)$$\mathrm{PC}\_\mathrm{Error}=10\log_{10}\left(\frac{3p^{2}}{max\left(e_{B,A}^{\mathrm{Dx}},\;e_{A,B}^{\mathrm{Dx}}\right)}\right),$$

In Equation (1), $e_{B,A}^{\mathrm{Dx}}\left(i\right)$ represents the point to point error in a full point cloud or error vector to a normal direction, and p is the peak constant value for the measured point cloud. A and B are the referenced and tested point cloud.

The compression ratio of each system was also analyzed. As introduced in Section 4.2, the 3D-HEVC coding system has three compression views. We considered one-third of the compressed size which was intended to be evaluated. The three systems were evaluated based on the final NR images. The average value was obtained from the NR images at three different depths. The PSNR and SSIM values were calculated by comparing the NR image from the original source and the reconstructed source data. Furthermore, 3D-HEVC and V-PCC coding systems were compared based on the original and reconstructed RGBD images, respectively. Both results were calculated from the average PSNR and SSIM of the depth and RGB images. These values were used to evaluate the source RGBD compression distortions.

## 5. Analyses of the Results

The analysis of the results includes a subjective evaluation of NR visual quality and an objective evaluation of PSNR and SSIM. Intermediate results on the PSNR of RGBD and PC_Error of point cloud data have also been evaluated to improve the understanding of the generation of the distortion. Furthermore, the compression ratio and encoding time are also important for the selection of three different codecs. The overall performance analysis and the limitations of the proposed method are presented at the end of this section.

### 5.1. Experimental Result

#### 5.1.1. Evaluation of Numerical Reconstruction

Figure 7 shows the NR images of Mario. We selected images with a compressed size of approximately 500 kbits. The bit rate for each system was chosen from the common test conditions of the respective codecs according to Ext_R6 for 3D-HEVC, Ext_Ra for HEVC, and R4 for V-PCC. By reviewing these images, we observed that even the RGB data were resampled into YUV 4:2:0 in the 3D-HEVC and V-PCC coding systems. However, color distortion was not obvious. The structural details are clearly visible in these images, and no obvious shape distortion was observed in the HEVC and V-PCC results. Distortion of one slit and slight color changes can be observed for the far depth, which is located on the hat part of the Mario figure. However, owing to the direct loss of hologram data, the NR images generated from the HEVC system have noticeable background noise.

The HEVC system takes hologram data directly into it, which contain a lot of high-frequency information that is not typically present in camera-captured lifetime data. This high-frequency information loss can cause noticeable background noise in the reconstructed images. These artifacts can also contribute to the background noise in the reconstructed images.

The Pororo NR images have a compressed source size of 250 kbits approximately. The NR images shown in Figure 8 were generated from the source at the rate point Ext_R6 on the 3D-HEVC coding system, Ext_Ra on HEVC, and R5 on the V-PCC. The results also showed that HEVC had the highest background noise intensity. However, depth loss can be observed on the helmet of Pororo with the 3D-HEVC system, and eye shape distortion can also be observed with V-PCC. A slight color change in the yellow component was also observed in these two results.

The NR images of the Cube data are shown in Figure 9. We selected images with a compressed size of approximately 400 kbits. The rates for each system were Ext_R8 for 3D-HEVC, R4 for HEVC, and R5 for V-PCC. An obvious depth loss can seen observed for the 3D-HEVC system, and V-PCC has the best visual quality.

#### 5.1.2. PSNR Evaluation

The PSNR results for each system are shown in Figure 10. The results show that both 3D-HEVC and V-PCC, apart from HEVC, have limitations with respect to the PSNR value. The color space conversion between RGB and YUV420 is the main reason for this distortion. Although HEVC showed a higher PSNR for four of the six rate points, the visual quality was comparatively low because of background noise.

Even though we extended the rate to obtain a higher quality point for comparison with other codecs, the highest encoding-configuration-generated 3D-HEVC bitstream size could not catch up with the HEVC encoded size. Additionally, the results showed that extended V-PCC rate points can provide a higher reconstruction quality than the rate points in the CTC.

In contrast to the objective evaluation results, in the subjective NR rendering of Pororo in Figure 11, the V-PCC result is the most ideal, the 3D-HEVC result has a loss in depth value, and the HEVC result has comprehensive noise.

#### 5.1.3. SSIM Evaluation

The SSIM results for each system are shown in Figure 12. The 3D-HEVC coding system had the lowest SSIM values compared to the other two systems. The V-PCC coding system yielded better SSIM results than the 3D-HEVC system when using a high-quality rate point such as Ext_R8. The color space conversion between RGB and YUV420 was the main reason for this distortion. Even HEVC yielded a better SSIM evaluation outcome in the medium-high range of extended rate points, although the objective quality was affected by the overall noise. For Pororo and Cube sequences, the V-PCC-generated bitstream size at the high end of the chart approaches the HEVC one, but the SSIM value is only 0.6 times that of HEVC.

In Figure 13, showing a subjective NR rendering of ‘Cube’, the V-PCC output is the most preferable. On the other hand, the 3D-HEVC display exhibits a diminished depth perception, and the HEVC output shows pervasive noise.

#### 5.1.4. Compression Ratio

The bar charts shown in Figure 14 are the quartile numbers of the compression ratio statistics. The circle outside of the bar chart is an outlier. The 3D-HEVC coding system yielded the largest compression ratio, ranging from 400 to 1000 times the maximum. However, the V-PCC coding system can provide a compression ratio in the range of 100 to 200.

The distribution of Mario data shows a larger compression ratio range. This could mean that a larger image size may express the performance potential to efficiently compress larger hologram images than median sized images. This is because larger images have more redundancy, which can be exploited by the coding algorithm to achieve a higher compression ratio.

#### 5.1.5. Evaluation of Intermediate RGBD Comparison

The PSNR results for the intermediate RGBD in Figure 15 show that V-PCC has a lower value by approximately 10 dB than that of 3D-HEVC. However, the SSIM results in Figure 16 show that both the 3D-HEVC and V-PCC coding systems have a high rating of 0.8. In particular, the V-PCC system could achieve an SSIM of 0.9. This means that the V-PCC system performs better for geometric compression.

#### 5.1.6. Evaluation of PC_Error

PC_Error can help evaluate the performance of V-PCC compression results. The PC_Error results in Figure 17 show the geometry and color quality. The Pororo sequence exhibits a slightly worse performance with respect to color loss, and the Cube sequence exhibits a worse geometry quality at a lower bitrate (e.g., R1 and R2) than the other two sequences. The results show that V-PCC performs better on larger CGH datasets. The PC_Error results for these three datasets lead to similar PSNR outcomes associated with the V-PCC processes of their CTC point cloud data [20].

#### 5.1.7. Encoding Time

The encoding time of the codec is also particularly important for the coding system because computational complexity is an essential issue for evaluation. The encoding time of each coding system was recorded on a PC with an Intel i7-6600K processor at 4.4 GHz with 16 GB of memory. The codec for each system was compiled using Microsoft Visual C++ 2017 in a Windows 10 environment. The range of the five-time average encoding time is shown in Figure 18 using bar charts. Compared with HEVC, 3D-HEVC and V-PCC are more time consuming. The encoding processes, such as inter-view prediction in 3D-HEVC and patch generation in V-PCC, require additional time. In this experiment, the HEVC coding system achieved the shortest coding times.

Additionally, an inconsistency in the encoding time of V-PCC and 3D-HEVC for the ‘Cube’ has been observed. The encoding time of V-PCC is higher than that of 3D-HEVC. The 3D-HEVC encoder may try multiple combinations of coding tools, which may contribute to the observed differences in performance. Some other early determination algorithms may also speed up the encoding process.

### 5.2. Performance Analysis

Although the subjective results of 3D-HEVC and V-PCC were lower than those of HEVC, the objective review of the NR images remained within an acceptable quality range. In the low bitrate range, 3D-HEVC and V-PCC maintained a low color loss after compression. This is because these two systems use color information from geometric structure information. Furthermore, the noise caused by lossy distortion was less in these two systems than in HEVC.

Based on the obtained results, the 3D-HEVC coding system can be used for compression, wherein the geometric structure is less important. This can benefit from the higher compression ratio of the 3D-HEVC scheme. For example, if you are compressing a video of a static hologram scene, the geometric structure is less dynamic than the color information. In this case, 3D-HEVC can achieve a higher compression ratio without sacrificing too much quality.

The V-PCC coding system yields a better coding performance for high-quality recordings than the other two methods. In addition, geometric details were coded efficiently using the V-PCC scheme. This makes V-PCC a good choice for compressing videos with a lot of detail, such as holograms with dynamic actions.

In the current experiment, an analysis with a limited view and frame number conditions were conducted. Further studies, such as on compression with multiple views and frames, need to be considered. For example, it would be interesting to see how the different codecs perform when compressing videos with a lot of views or a lot of frames. This would help to determine the best codec for specific applications. Overall, the results of this study suggest that 3D-HEVC and V-PCC are both viable options for compressing 3D videos. The choice of which codec to use will depend on the specific application and the desired quality level.

## 6. Conclusions

In this study, we designed and implemented 3D-HEVC, HEVC, and V-PCC coding systems and evaluated their objective and subjective results in RGBD hologram data compression. By reviewing the objective visual quality of the compression performances, 3D-HEVC and V-PCC were found to be reasonable choices for RGBD source CGH data compression. These compression methods can be used in future development.

Some artifacts caused by the 3D-HEVC and V-PCC coding systems indicate that rate distortion optimization is still needed. Problems related to the bitrate optimization of the geometry structure information or color information may cause differences in the results. In future studies, we plan to perform a more detailed analysis of the coding tools used in these codecs and investigate the potential of using the Versatile Video Coding (VVC) codec for RGBD hologram data compression. We also plan to explore the compression of multiple frames of dynamic holograms or multiview RGBD, which could benefit from these designs.

## Figures and Tables

**Figure 1 sensors-23-07684-f001:**
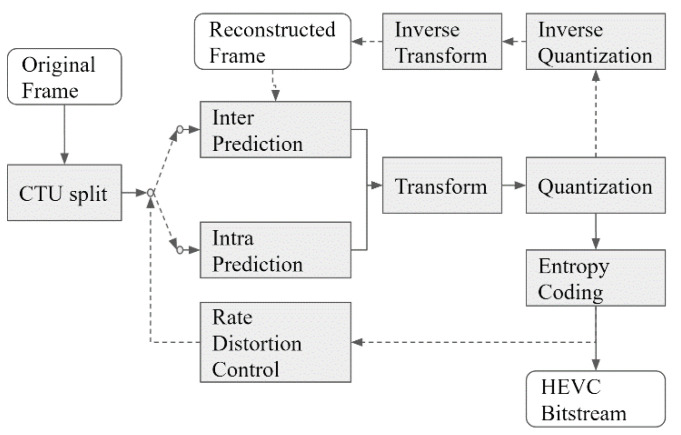
High-efficiency video coding (HEVC) encoder diagram.

**Figure 2 sensors-23-07684-f002:**
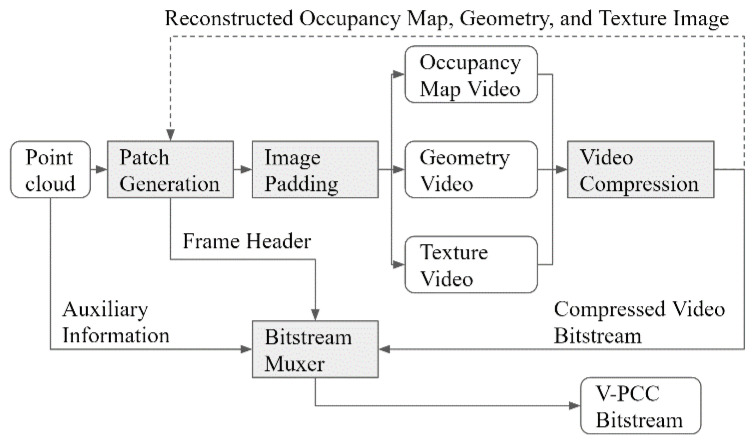
Video-based point cloud compression (V-PCC) encoder diagram.

**Figure 3 sensors-23-07684-f003:**
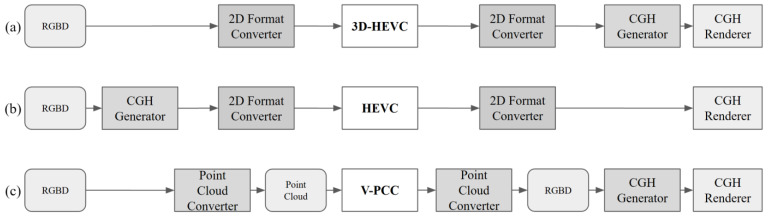
Diagram of coding system design. (**a**) 3D-HEVC-based coding system; (**b**) HEVC-based system; (**c**) V-PCC-based coding system.

**Figure 4 sensors-23-07684-f004:**
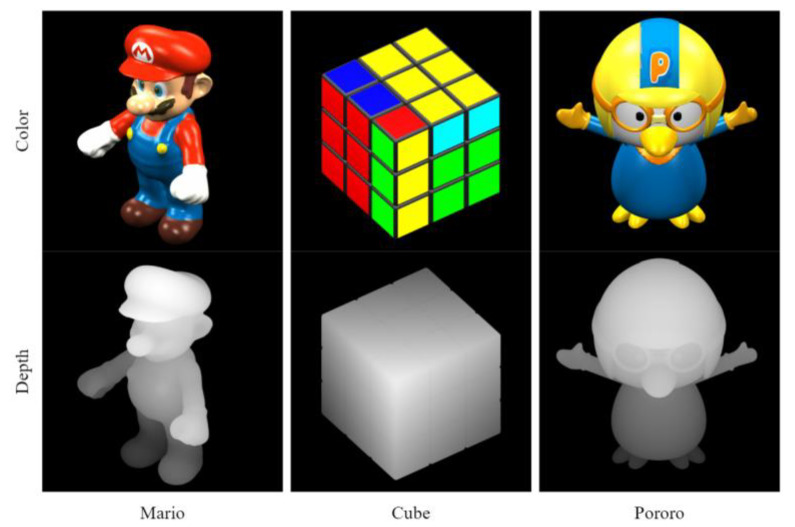
Original RGBD test images.

**Figure 5 sensors-23-07684-f005:**
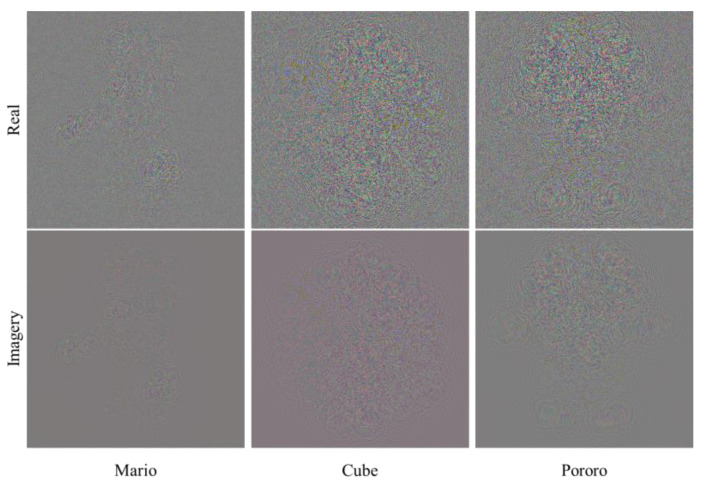
Hologram images generated from test images.

**Figure 6 sensors-23-07684-f006:**
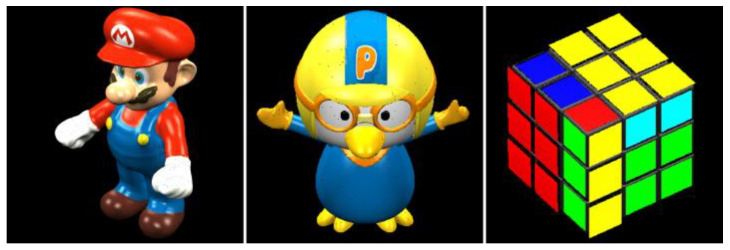
Rendering images of point cloud data generated from test images.

**Figure 7 sensors-23-07684-f007:**
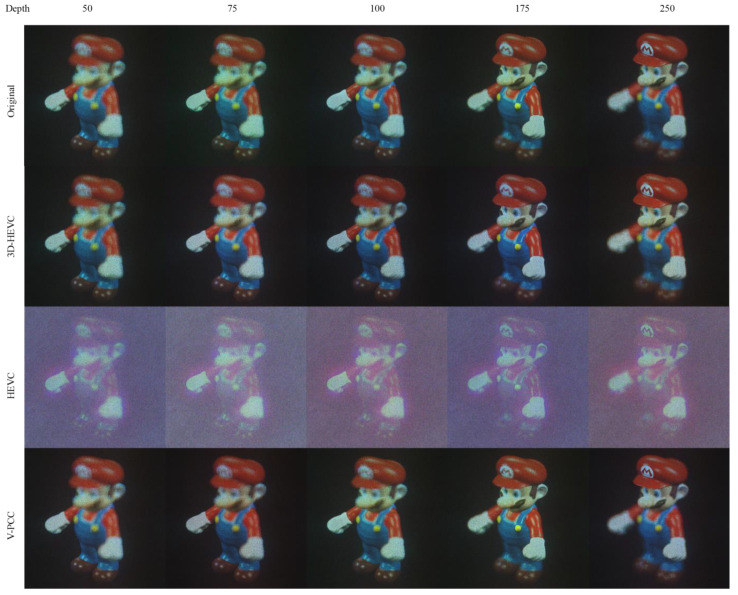
NR result for Mario.

**Figure 8 sensors-23-07684-f008:**
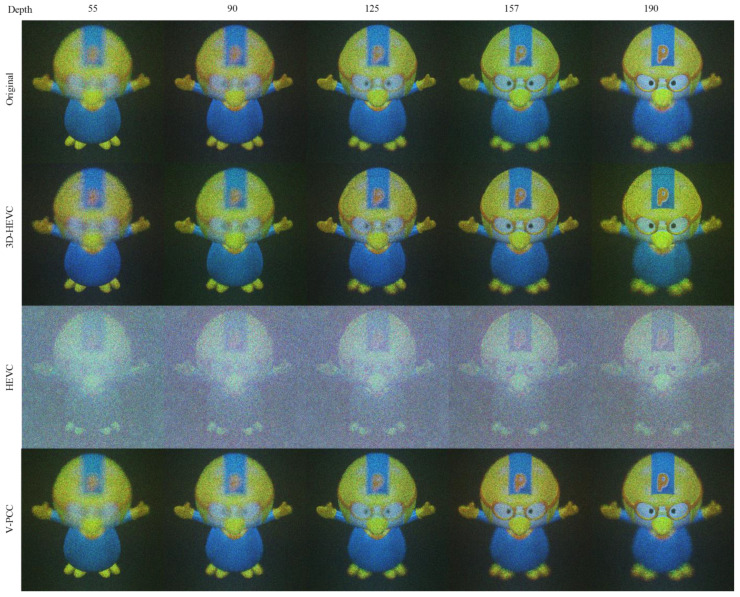
NR result for Pororo.

**Figure 9 sensors-23-07684-f009:**
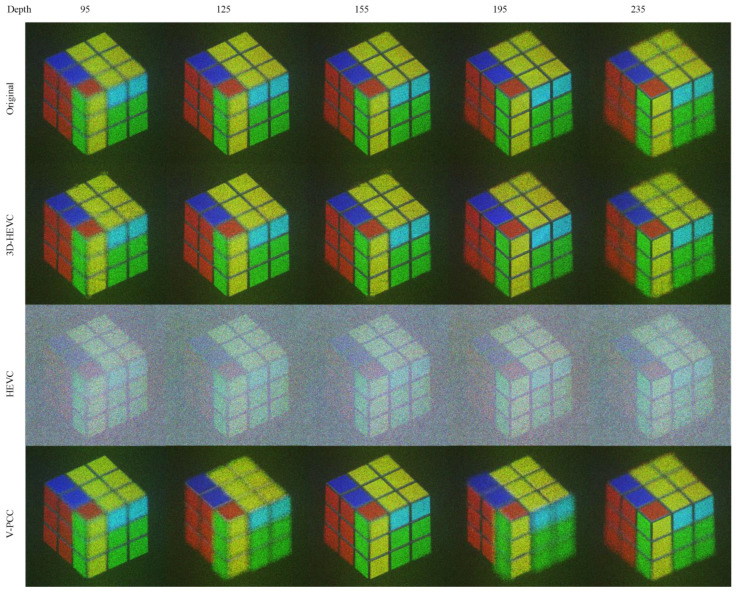
NR result for Cube.

**Figure 10 sensors-23-07684-f010:**
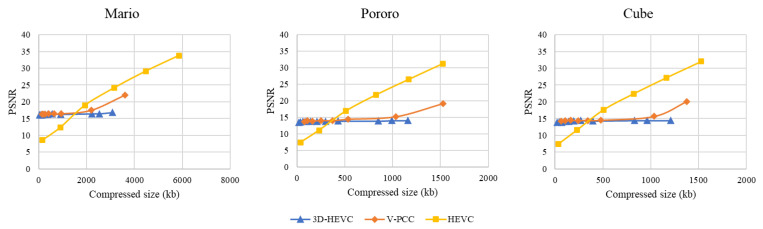
Peak signal-to-noise ratio (PSNR) evaluation.

**Figure 11 sensors-23-07684-f011:**
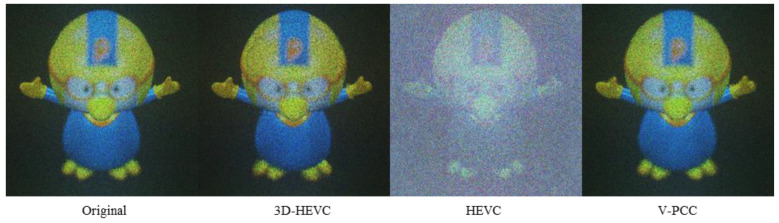
NR result for Pororo. (Depth 125).

**Figure 12 sensors-23-07684-f012:**
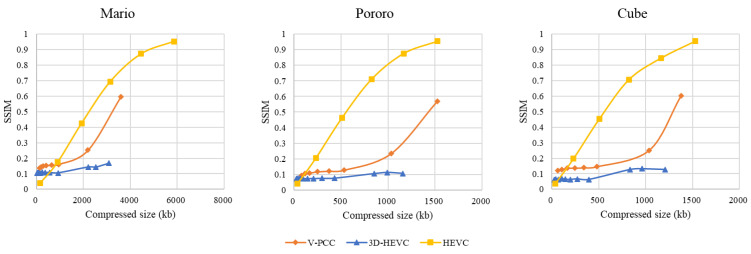
Structure similarity (SSIM) evaluation.

**Figure 13 sensors-23-07684-f013:**
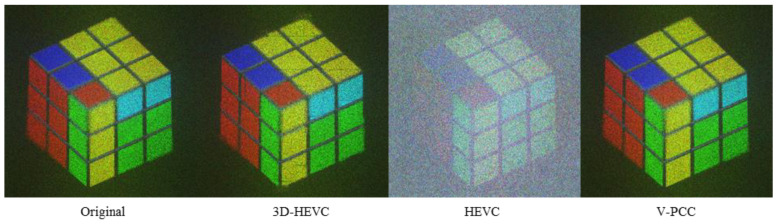
NR result for Cube (Depth 155).

**Figure 14 sensors-23-07684-f014:**
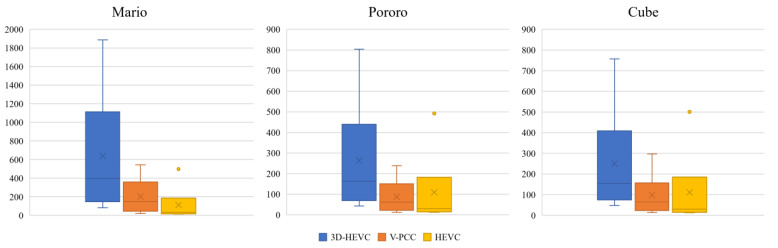
Compression ratio outcome (The circle outside of the bar chart is an outlier. The circle means point fall significantly outside the expected range given the rest of the data).

**Figure 15 sensors-23-07684-f015:**
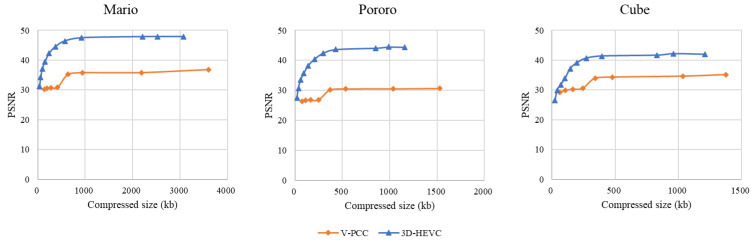
PSNR for intermediate RGBD.

**Figure 16 sensors-23-07684-f016:**
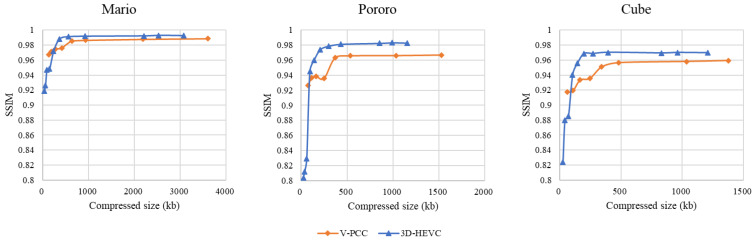
SSIM for intermediate RGBD.

**Figure 17 sensors-23-07684-f017:**
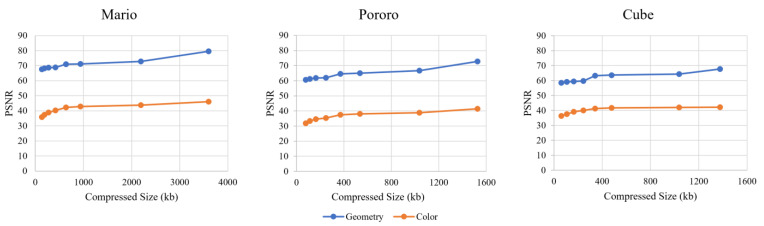
Intermediate PC_Error outcomes of V-PCC system.

**Figure 18 sensors-23-07684-f018:**
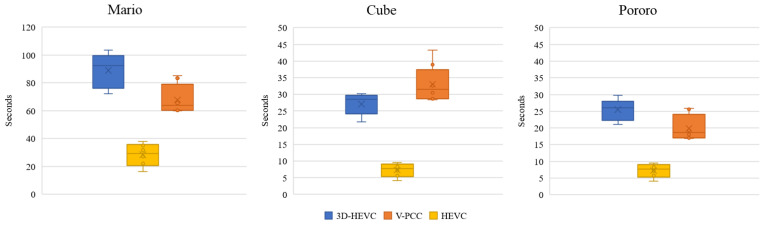
Encoding time outcomes of each codec.

**Table 1 sensors-23-07684-t001:** Existing holography compression methods.

Category	Method	Target Data
legacy image codec	JPEG [4]	hologram data
	JPEG 2000 [5]	
	HEVC [6]	
hologram pattern compression	Fresnel transform based JPEG 2000 [7]	transformed hologram data
	Fpzip [8]	
	Gabor’s dictionary matching pursuit [9]	
	wave atom coding [10]	
neural networks and deep learning	deep neural network [13]	reconstructed hologram data
	deep convolutional neural network [14]	

**Table 2 sensors-23-07684-t002:** Rate points of the 3D-HEVC coding system.

Rate Point	R1	R2	R3	R4	R5	Ext R6	Ext R7	Ext R8	Ext R9	Ext R10	Ext R11
Color QP	45	40	35	30	25	20	15	10	5	1	1
Geometry QP	48	45	42	39	34	29	24	19	15	9	5

**Table 3 sensors-23-07684-t003:** Rate points of the HEVC coding system.

Rate Point	Ext Rb	Ext Ra	R1	R2	R3	R4
Color QP	47	42	37	32	24	22

**Table 4 sensors-23-07684-t004:** Rate points of the V-PCC coding system.

Rate Point	R1	R2	R3	R4	R5	Ext R6	Ext R7	Ext R8
Color QP	42	37	32	27	22	17	12	7
Geometry QP	32	28	24	20	15	12	8	4
Occupancy Map Precision	4	4	4	4	2	2	2	2

**Table 5 sensors-23-07684-t005:** Rate point of the V-PCC coding system.

Test Image	Depth				
Mario	50	75	100	125	250
Pororo	55	95	125	157	190
Cube	95	125	155	190	235

## Data Availability

These experimental datasets are the result of the “table-top holographic display terminal R&D project” under the Giga Korea Project. The dataset underlying the results presented in this paper is not publicly available at this time but may be obtained from the authors upon reasonable request. The MATLAB script for point cloud to RGBD conversion is available in [26].

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
