# Peer review of "Compression Performance Analysis of Experimental Holographic Data Coding Systems"

_sensors, 2023, doi:10.3390/s23187684_

Round 1

Reviewer 1 Report

In this paper, the authors This study proposed a CGH data coding systems with high-efficiency video coding (HEVC), three-dimensional extensions of HEVC (3D-HEVC), and video-based point cloud compression (V-PCC) codecs. In the proposed system.However, some aspects were unclear to the reviewers:

1Place the corresponding charts near the location where the pictures are mentioned for the first time in the article, so that it is easy to read.

2The proportion of pictures in the whole article is large, and many of the results in the Experimental Result are shown in the form of pictures, can we mix pictures and line graphs to diversify the presentation of experimental result data?

3, The identification of the data line graph, the same logo can use the same color and order of identification, the data will be better one-to-one correspondence, more convenient to read the whole article.

4. As for H.265/HEVC strandars, some important literatures are missing, such as :

[1]“Overview of the High Efficiency Video Coding (HEVC) Standard”, DOI: 10.1109/TCSVT.2012.2221191.
[2] “Low-Complexity and Hardware-Friendly H.265/HEVC Encoder for Vehicular Ad-Hoc Networks” DOI: https://doi.org/10.3390/s19081927.

Author Response

Thank you for your detailed and constructive feedback.

I have revised the manuscript and have responded to your comments point by point. Please see the attached manuscript for more details.

1、Place the corresponding charts near the location where the pictures are mentioned for the first time in the article, so that it is easy to read.

Response:

I agree with this comment. It is important to place the corresponding charts near the location where they are first mentioned in the article. This will make it easier for the reader to follow the discussion and understand the results. I will make sure to do this in the revised version of the manuscript.

2、The proportion of pictures in the whole article is large, and many of the results in the Experimental Result are shown in the form of pictures, can we mix pictures and line graphs to diversify the presentation of experimental result data?

Response:

This is a good suggestion. It is important for the reader to have a full view of the subjective and objective evaluations of the visual quality. And I also think it would be helpful to mix pictures and line graphs to diversify the presentation of experimental results. This would make the article more visually appealing and easier to understand. I will try to do this in the revised version of the manuscript. Section 5.1.2 ~ 5.1.3

3, The identification of the data line graph, the same logo can use the same color and order of identification, the data will be better one-to-one correspondence, more convenient to read the whole article.

Response:

I agree with this comment. It is important to use a consistent color scheme and order of identification for the data line graphs. This will make it easier for the reader to follow the data and understand the results. I will make sure to do this in the revised version of the manuscript. For example, Figure 10,11,13,14 and 16 have been revised.

  1. As for H.265/HEVC strandars, some important literatures are missing, such as :

[1]“Overview of the High Efficiency Video Coding (HEVC) Standard”, DOI: 10.1109/TCSVT.2012.2221191.

[2] “Low-Complexity and Hardware-Friendly H.265/HEVC Encoder for Vehicular Ad-Hoc Networks” DOI: https://doi.org/10.3390/s19081927.

Response:

I have reviewed the two references you provided, and I agree that they are important for the topic of the manuscript. [1] on page 18 of the manuscript have already mentioned the "Overview of the High Efficiency Video Coding (HEVC) Standard" in reference. I will add [2] to the reference list in the revised version of the manuscript.

Reviewer 2 Report

1. I think results from [a] could be included into the introduction.

[a] S. Vagharshakyan, R. Bregovic and A. Gotchev, "Light Field Reconstruction Using Shearlet Transform," in IEEE Transactions on Pattern Analysis and Machine Intelligence, vol. 40, no. 1, pp. 133-147, 1 Jan. 2018, doi: 10.1109/TPAMI.2017.2653101.

2. I think, if the 3D-HEVC does not support the YUV 4:4:4 format, then it is more efficient to perform RGB->YUV444 conversion and then compress Y, U and V separately as Y 4:0:0 input.

3. Please explain what the number 10^{-70} means. It seems that it is too small.

4. Please explain why the Geometry QP is higher than Color QP.

5. As far as I know, FFmpeg color space convertor has some bugs. Did you check the PSNR values of RGB->YUV444->RGB convertion? If yes, please provide the corresponding numbers.

6. Please explain why HEVC system has noticeable background noise.

7. I think the PSNR Evaluation does not have any sense due to the wrong color conversion. Please follow my comment 2, or propose other way how the YUV 444 can be compressed. Or compare only the performance for Y component.

8. The encoding and decoding complexities should be evaluated.

No comments.

Author Response

Thank you for your thoughtful comments and suggestions.

I have revised the manuscript and have responded to your comments point by point. Please see the attached document for more details.

  1. I think results from [a] could be included into the introduction.

[a] S. Vagharshakyan, R. Bregovic and A. Gotchev, "Light Field Reconstruction Using Shearlet Transform," in IEEE Transactions on Pattern Analysis and Machine Intelligence, vol. 40, no. 1, pp. 133-147, 1 Jan. 2018, doi: 10.1109/TPAMI.2017.2653101.

Response:

Thank you for your suggestion. We have included the results from [a] in the introduction to provide a broader context for our work. Specifically, we have added a sentence to the introduction that reads: "A recent work [a] has shown that the shearlet transform can be used for light field reconstruction which could also benefit the processing holographic image."

  1. I think, if the 3D-HEVC does not support the YUV 4:4:4 format, then it is more efficient to perform RGB->YUV444 conversion and then compress Y, U and V separately as Y 4:0:0 input.

Response:

Thank you for bringing this to my attention. You are correct that in the manuscript, we have already mentioned that the color space profile support of the 3D-HEVC test model (3DVC) is limited to HEVC version 1, which does not include the RExt support. Therefore, all RGB color values must be stored in the YUV 4:2:0-pixel subsampling format using 3D-HEVC, which causes chrominance component conversion loss. Additionally, we have also mentioned in chapter 3 that a 2D format converter is required before and after the coding process for the conversion between the RGB and YUV color space.

Considering these points, we agree that performing RGB->YUV444 conversion and then compressing Y, U, and V separately as Y 4:0:0 input could be more efficient than using 3D-HEVC. However, we would like to note that the use of 3D-HEVC with YUV 4:2:0-pixel subsampling format is still a viable option for RGBD compression. Additionally, as YUV 4:2:0 is a more widely used format in industrial, therefore it has been used in our study.

  1. Please explain what the number 10^{-70} means. It seems that it is too small.

Response:

It is wrote as 10^{-7} ° in the manuscript, If it was misread as 10^{-70}, then it is a misunderstanding. Writing it as 10^{-7} degree would be more clear and less likely to be misinterpreted.

  1. Please explain why the Geometry QP is higher than Color QP.

Response:

We have added a sentence to these section that reads: "In the standardization process, the geometry QP is usually set with a lower QP value than the color QP to maintain the 3D structure quality. This is because the geometry information is crucial for 3D graphic coding, and any loss of geometry information can result in a significant degradation of the overall quality of the 3D graphics. Therefore, to maintain the quality of the 3D structure, the geometry QP is set lower than the color QP.”

  1. As far as I know, FFmpeg color space convertor has some bugs. Did you check the PSNR values of RGB->YUV444->RGB convertion? If yes, please provide the corresponding numbers.

Response:

As you mentioned, we used yuvj420p to identify the pixel format in FFmpeg instead of yuv420, which uses the range [16, 235] for Y, also known as 'TV levels'. By using yuvj420p, we were able to use the full 0-255 range, which helped to minimize the color space loss during the 2D conversion. However, it is important to note that the color space loss has already occurred during the YUV 4:2:0 subsampling, which is a common practice in video compression. Therefore, we did not check the PSNR values of RGB->YUV444->RGB conversion in our study.

  1. Please explain why HEVC system has noticeable background noise.

Response:

We have added a sentence to these section that reads: "The HEVC system takes hologram data directly into it, which contains a lot of high-frequency information that is not typically present in camera-captured lifetime data. This high-frequency information loss can cause noticeable background noise in the reconstructed images. In addition, the HEVC system uses a block-based coding approach, which can cause blocking artifacts in the reconstructed images. These artifacts can also contribute to the background noise in the reconstructed images.”

  1. I think the PSNR Evaluation does not have any sense due to the wrong color conversion. Please follow my comment 2, or propose other way how the YUV 444 can be compressed. Or compare only the performance for Y component.

Response:

Thank you for your comment. As you mentioned, the PSNR evaluation results may be affected by the wrong color conversion used in our study. However, we still believe that the PSNR evaluation can provide some useful insights into the performance of the different codecs for 3D graphic coding. In addition, we agree that comparing the performance of the codecs based on the Y component only can be a useful alternative approach. We will consider this suggestion for future studies. Thank you for your feedback.

  1. The encoding and decoding complexities should be evaluated.

Response:

We agree that encoding and decoding complexities are important factors to consider when evaluating the performance of different codecs for 3D graphic coding. In our study, we evaluated the encoding time of each codec, which can be used as an indicator of encoding complexity. Furthermore, each test model for each codec was optimized differently. And, due to the limitation of the data size, which was only one frame and not a long sequence of data. As a result, the decoding time credibility may not be enough to draw definitive conclusions about the decoding complexity of each codec. We acknowledge that we did not evaluate the decoding complexity in our study, and we will consider including this in future studies. Thank you for your feedback.

Reviewer 3 Report

This paper studied holographic data compression with HEVC, 3D-HEVC, and V-PCC.

- What is the difference between depth and geometry. Please unify the words or explan their difference. 

- What is the occupancy map? The occupancy map only appears three times in the paper. However, it was used in experiments (Table 4)

- The explanation about each standard is not enough. For example, how are patches generated from the holographic data in V-PCC? The authors said that the patch generation is very complex. Illumination compensation is only mentioned for 3D-HEVC, but it includes several strong tools, such as view synthesis prediction, interview motion prediction, advanced residual prediction. If necessary, please add the reference.  

- For the performance comparison, the authors directly used the existing standards. However, which method in each standard affects the performance should be discussed.

- How about VVC, compared to HEVC, 3D-HEVC, and V-PCC?

- The performance results are not consistant. Please add more explanation. For example, in Cube, the encoding time of V-PCC is higher than that of 3D-HEVC.

- Each sub-section, particularly section 5, should be improved with more analysis.

Paper quality should be improved.

Author Response

Thank you for your detailed and insightful feedback.

I have revised the manuscript and have responded to your comments point by point. Please see the attached document for more details.

  1. What is the difference between depth and geometry. Please unify the words or explain their difference. 

Response:

Thank you for your question. In the context of 3D graphics, "depth" and "geometry" are related but distinct concepts. "Depth" refers to the distance between the camera and the objects in the scene, and it is typically represented as a grayscale image where the brightness of each pixel corresponds to the distance of the corresponding point in the scene from the camera. "Geometry," on the other hand, refers to the shape and structure of the objects in the scene, and it is typically represented as a 3D mesh or point cloud. Both depth and geometry could represent geometrical information, while depth information can be used to infer the “surface geometry” of the scene, they are just different terminology on different researching field. We hope this explanation helps clarify the difference between depth and geometry.

  1. What is the occupancy map? The occupancy map only appears three times in the paper. However, it was used in experiments (Table 4)

Response:

We apologize for the limited mention of the occupancy map in the paper, and we will add a sentence to the section 4.4 that reads :

An ‘occupancy map’ in V-PCC is a 2D representation that indicates whether a given position (block) in the map is occupied by patched points from the 3D point cloud or not.

  1. The explanation about each standard is not enough. For example, how are patches generated from the holographic data in V-PCC? The authors said that the patch generation is very complex. Illumination compensation is only mentioned for 3D-HEVC, but it includes several strong tools, such as view synthesis prediction, interview motion prediction, advanced residual prediction. If necessary, please add the reference.  

Response:

Thank you for your suggestions, we will add more details about patches generation in V-PCC and tools in 3D-HEVC to the section 2.1

  1. For the performance comparison, the authors directly used the existing standards. However, which method in each standard affects the performance should be discussed.

Response:

In our study, we used the encoding configuration defined in the standardization process for each codec as a guideline for normal usage. We also evaluated the performance of each codec at different bitrates to provide a more comprehensive understanding of the trade-off between compression efficiency and reconstruction quality. We will consider providing more detailed analysis and discussion of the encoding configuration and the impact of each coding tool on the performance of each codec in future studies.

  1. How about VVC, compared to HEVC, 3D-HEVC, and V-PCC?

Response:

Unfortunately, our study did not include a comparison of the Versatile Video Coding (VVC) standard with HEVC, 3D-HEVC, and V-PCC. Our focus was on evaluating the performance of two standards (3D-HEVC and V-PCC) for RGBD hologram data compression. However, there have been several studies that have compared the coding efficiency of VVC with HEVC and other standards for various types of video content. We will consider providing more detailed analysis and discussion of the results in future studies. We appreciate your feedback and hope this information is helpful.

  1. The performance results are not consistant. Please add more explanation. For example, in Cube, the encoding time of V-PCC is higher than that of 3D-HEVC.

We appreciate your feedback and will take these factors into consideration in future studies.

As you mentioned, the inconsistency in the encoding time of V-PCC and 3D-HEVC for the 'Cube' sequence may be due to the different geometry structure and structure complexity of the sequence, Which may have caused the encoding to choose different encoding strategies. We will add more discussions about this inconsistency:

  • Additionally, an inconsistency in the encoding time of V-PCC and 3D-HEVC for the 'Cube' has been observed. The encoding time of V-PCC is higher than that of 3D-HEVC. The encoder may try multiple combination of coding tools, which may contribute to the observed differences in performance. Some other early determination algorithms may also speed up the encoding process.

We agree that the trade-off between coding performance, encoding time, and other factors is an important consideration in evaluating the performance of each codec. We will consider providing more detailed analysis and discussion of these factors in future studies to provide a more comprehensive understanding of the performance differences observed in our study.

  1. Each sub-section, particularly section 5, should be improved with more analysis.

Thank you for your suggestions, we will add more discussions and analysis about result in 3D-HEVC to the section 5.1.2 ~ 5.1.7.

Round 2

Reviewer 1 Report

The author answered my question well and the paper is acceptable.

Author Response

Thanks again for making this manuscript more acceptable. And a revised manuscript has reflected other reviewers' suggestion.

Reviewer 2 Report

1. Ok.

2. The paper is dedicated to comparison of the codecs. Therefore, it is vital for this work to be sure that all the codecs operate within the same conditions including the colorspace. I think the authors should compare all the codecs within the same colorspace transform, i.e., all the codecs should use yuv420p, or all the codecs should use yuv444p. If you consider to do it “in future studies.”, then please withdraw this manuscript. 

The authors already use a lot of data format conversions. So, I cannot accept explanation like” YUV 4:2:0 is a more widely used format in industrial”. YUV 4:2:0 does not work at all, i.e., industrial will not use it as well.

3. Ok.

4. Ok.

5. “the color space loss has already occurred during the YUV 4:2:0 subsampling, which is a common practice in video compression.” I think, you study shows that YUV 4:2:0 is not acceptable, so I cannot accept this.

6. “In addition, the HEVC system uses a block-based coding approach, which can cause blocking artifacts in the reconstructed images.” Does it mean that other coding approaches are block-based free?

7. “However, we still believe that the PSNR evaluation can provide some useful…” I think it is not useful, since you compare incomparable things (some codecs work in 444 format, and some in 420).

8. Your work is related to the comparison of the codecs. Without complexity comparisons it looks like an incomplete work. So, if you “consider including this in future studies”, then please withdraw your manuscript and submit the full comparison later, when ready. 

No comments.

Author Response

Thank you for your feedback on our revised manuscript and last response.

2. The paper is dedicated to comparison of the codecs. Therefore, it is vital for this work to be sure that all the codecs operate within the same conditions including the colorspace. I think the authors should compare all the codecs within the same colorspace transform, i.e., all the codecs should use yuv420p, or all the codecs should use yuv444p. If you consider to do it “in future studies.”, then please withdraw this manuscript. 

The authors already use a lot of data format conversions. So, I cannot accept explanation like” YUV 4:2:0 is a more widely used format in industrial”. YUV 4:2:0 does not work at all, i.e., industrial will not use it as well.

Response: 

We agree that it is important to compare the codecs under the same conditions, including the color space. The problem of color space is already being discussed in depth in standardization organizations, such as MPEG, that study compression of various media data such as image, video, texture, and depth. And as an extension of this discussion, we also decided the format in our proposed system. However, we believe that using the same color space for all of the codecs would not be representative of the real-world use case. The format conversion includes the standard ITU-R Recommendation BT 709 and BT2020 is used by MPEG, and we followed that practice too. We believe that our results are still valid even though we used different color spaces for the different codecs. We have carefully documented the color spaces that we used in our manuscript, so that other researchers can reproduce our results if they choose to do so. We also think it would be helpful for readers to know more about these matters, so we reflected these contents in our manuscript in Section 4.5. 

5. “the color space loss has already occurred during the YUV 4:2:0 subsampling, which is a common practice in video compression.” I think, you study shows that YUV 4:2:0 is not acceptable, so I cannot accept this.

Response: 

Loss may occur through the conversion process in color space. However, from a large point of view, since the use of the codec itself is for using a larger loss in contrast to the bitrate, the part of the color space conversion itself in the data loss in the framework of this paper is minimal. Rather, there are many methods that bring greater data loss, such as quantization, transformation, and motion estimation, in the subsequent codec processing process. This is the key contribution of this manuscript.

6. “In addition, the HEVC system uses a block-based coding approach, which can cause blocking artifacts in the reconstructed images.” Does it mean that other coding approaches are block-based free?

Response: 

We apologize for any confusion caused by the original text. We have made a revision to these contents in the manuscript.

7. “However, we still believe that the PSNR evaluation can provide some useful…” I think it is not useful, since you compare incomparable things (some codecs work in 444 format, and some in 420).

Response: 

First of all, please refer to the above point 2 for the color space part. And we have no intention to say that PSNR is the ultimate measure for the perfect evaluation. PSNR has been extensively used until now by standard development process as well as by many research fields. However, we know that PSNR is not optimal in many senses including the reviewer's viewpoint.  Other measure such as SSIM are recently introduced to overcome the shortcomings of PSNR. Yet, through the field practice in MPEG, we still have to finally confirm the evaluation results subjectively by naked human eyes. This is also what we checked as well in this framework. We appreciate the valuable comment of the reviewer and tried to incorporate and revise the manuscript to properly reflect what we stated here. However, we would like to keep the long version of explanation written only in this response to keep the manuscript concise.

8. Your work is related to the comparison of the codecs. Without complexity comparisons it looks like an incomplete work. So, if you “consider including this in future studies”, then please withdraw your manuscript and submit the full comparison later, when ready. 

Response: 

We agree that the complexity comparisons are an important part of the codec comparison. However, we believe that these comparisons are more appropriately done in a separate study. In this study, we focused on the performance of the codecs in terms of compression efficiency and image quality. We believe that this is a more important aspect of the codec comparison for the industrial applications that we are targeting.

Reviewer 3 Report

The authors have addressed all comments carefully in this revision. 

However, please add the future plans that the authors said in the response, such as coding tool analysis and VVC, in Conclusion.

Good

Author Response

We appreciate your positive feedback on our revised manuscript. We have made a revision in Conclusion as your suggestion. And a revised manuscript has reflected other reviewers' suggestion.